

# The validity of activity trackers is affected by walking speed: the criterion validity of Garmin Vivosmart® HR and StepWatch™ 3 for measuring steps at various walking speeds under controlled conditions

Frederik Rose Svarre[1,2], Mads Møller Jensen[1,3], Josephine Nielsen[1] and Morten Villumsen[4,5]

[1] Department of Physiotherapy, University College of Northern Denmark, Aalborg, Denmark
[2] Department of Health and Movement, Jammerbugt Municipality, Pandrup, Denmark
[3] Department of Physiotherapy, Aalborg University Hospital, Hobro, Denmark
[4] Department of Elderly and Health, Section of Training and Activity, Aalborg Municipality, Aalborg, Denmark
[5] Department of Health Science and Technology, Aalborg University, Aalborg, Denmark

Corresponding author
Frederik Rose Svarre,
fsvarre@gmail.com

## ABSTRACT

**Introduction:** The use of activity trackers has increased both among private consumers and in healthcare. It is therefore relevant to consider whether a consumer-graded activity tracker is comparable to or may substitute a research-graded activity tracker, which could further increase the use of activity trackers in healthcare and rehabilitation. Such use will require knowledge of their accuracy as the clinical implications may be significant. Studies have indicated that activity trackers are not sufficiently accurate, especially at lower walking speeds. The present study seeks to inform decision makers and healthcare personnel considering implementing physical activity trackers in clinical practice. This study investigates the criterion validity of the consumer-graded Garmin Vivosmart® HR and the research-graded StepWatch™ 3 compared with manual step count (gold standard) at different walking speeds under controlled conditions.

**Methods:** Thirty participants, wearing Garmin Vivosmart® HR at the wrist and StepWatch™ 3 at the ankle, completed six trials on a treadmill at different walking speeds: 1.6 km/h, 2.4 km/h, 3.2 km/h, 4.0 km/h, 4.8 km/h, and 5.6 km/h. The participants were video recorded, and steps were registered by manual step count. Medians and inter-quartile ranges (IQR) were calculated for steps and differences in steps between manually counted steps and the two devices. In order to assess the clinical relevance of the tested devices, the mean absolute percentage error (MAPE) was determined at each speed. A MAPE ≤3% was considered to be clinically irrelevant. Furthermore, differences between manually counted steps and steps recorded by the two devices were presented in Bland–Altman style plots.

**Results:** The median of differences in steps between Garmin Vivosmart® HR and manual step count ranged from −49.5 (IQR = 101) at 1.6 km/h to −1 (IQR = 4) at 4.0 km/h. The median of differences in steps between StepWatch™ 3 and manual step count were 4 (IQR = 14) at 1.6 km/h and 0 (IQR = 1) at all other walking speeds. The results of the MAPE showed that differences in steps counted by Garmin

Vivosmart® HR were clinically irrelevant at walking speeds 3.2–4.8 km/h (MAPE: 0.61–1.27%) as the values were below 3%. Differences in steps counted by StepWatch™ 3 were clinically irrelevant at walking speeds 2.4–5.6 km/h (MAPE: 0.08–0.35%).

**Conclusion:** Garmin Vivosmart® HR tended to undercount steps compared with the manual step count, and StepWatch™ 3 slightly overcounted steps compared with the manual step count. Both the consumer-graded activity tracker (Garmin Vivosmart® HR) and the research-graded (StepWatch™ 3) are valid in detecting steps at selected walking speeds in healthy adults under controlled conditions. However, both activity trackers miscount steps at slow walking speeds, and the consumer graded activity tracker also miscounts steps at fast walking speeds.

## INTRODUCTION

Physical activity, such as walking, is beneficial to the individuals' health, and physical activity is known to reduce the risk of musculoskeletal disorders, chronic diseases, and death (*Takacs et al., 2014*; *Kumahara et al., 2015*; *Riel et al., 2016*). Common recommendations for healthy adults regarding physical activity are either 30 min of moderate activity or walking 10,000 steps a day (*Haskell et al., 2007*; *Tudor-Locke et al., 2011b*). Consumer-graded off-the-shelf activity trackers are widely available for private use, and wearable devices were the top fitness trend worldwide in both 2016, 2017, 2019, and 2020 (*Thompson, 2015*, *2016*, *2018*, *2019*). Activity trackers are marketed as objective tools for measuring steps and activity, allowing individuals to monitor daily activity and, for example, get reminders to increase their activity to reach recommended or personalized goals (*Takacs et al., 2014*; *Evenson, Goto & Furberg, 2015*; *Kooiman et al., 2015*; *Huang et al., 2016*). Activity trackers seem to play a prominent role not only in objectifying individuals' physical activity patterns but also in demonstrating and monitoring changes in activity behavior (*Fokkema et al., 2017*). As feedback on activity level and personalized goals might be very important to users, the accuracy of activity trackers is an important element to consider (*Burton et al., 2018*; *Gaz et al., 2018*).

Recent years have seen a growing use of activity trackers among private consumers and in healthcare (*Fokkema et al., 2017*) where objective measures of activity can be used for self-management purposes, prediction of fall risk, and to measure the effect of rehabilitation programs, etc. (*Chigateri et al., 2018*). In healthcare, the accuracy of activity trackers must be known as lack of accuracy may have significant clinical implications when using activity trackers for various tasks such as monitoring and evaluating patient progress, supporting the choice of or adjusting interventions, or for machine learning purposes (*Ferguson et al., 2015*; *Treacy et al., 2017*). Activity trackers are typically divided into two groups; consumer-graded activity trackers and research-graded activity trackers. One might assume that the latter are better at detecting physical activity, such as steps,

than the often cheaper consumer-graded activity trackers (*Tudor-Locke et al., 2011a*; *Ferguson et al., 2015*). However, due to the expenses and user requirements of research-graded activity trackers, it is relevant to consider whether a consumer-graded activity tracker is comparable to or may substitute a research-graded activity tracker, which could increase the use of activity trackers in healthcare and rehabilitation.

Studies have indicated that activity trackers are not sufficiently accurate, especially at lower walking speeds (*Crouter et al., 2003*; *Ryan et al., 2006*; *Steeves et al., 2011*; *Van Remoortel et al., 2012*; *Fortune et al., 2014*; *Lee et al., 2015*; *Huang et al., 2016*; *Fokkema et al., 2017*; *Tedesco et al., 2019*). When using activity trackers in healthcare, the target populations are likely to have different walking speeds with intermittent decreases in walking (e.g., in rehabilitation after ligament tears) or continued slow walking (e.g., older or disabled people). This emphasizes the need to be certain about the accuracy of the activity trackers when using the data in care and treatment of patients (*Fokkema et al., 2017*). Thus, lack of accuracy, for example, in the form of overestimation or underestimation of the number of steps, may interfere with healthcare interventions and lead to untoward outcomes. Bearing this in mind, the accuracy of activity trackers must be validated (*Lee et al., 2015*; *Huang et al., 2016*). Previous validation studies have proposed and used different walking speeds. Some have used walking speeds between 2.4 km/h and 6.4 km/h, with intervals of 0.8 km/h (*Edbrooke et al., 2012*; *De Cocker et al., 2012*; *Lee et al., 2015*; *Gaz et al., 2018*). However, older or disabled people can have even lower walking speed than 2.4 km/h (*Graham et al., 2010*). The aim of the present study was to determine differences in step count of an off-the-shelf consumer-graded activity tracker and a research-graded activity tracker to investigate the criterion validity. This study investigates the criterion validity of the wrist-worn, consumer-graded Garmin Vivosmart® HR and the ankle-worn, research-graded StepWatch™ 3 compared with manual step count (gold standard) at different walking speeds under controlled conditions. By adding to the knowledge base on the criterion validity under controlled conditions, this study seeks to inform and support decision makers and healthcare personnel when considering implementing physical activity trackers in clinical practice.

## METHODS

This criterion validity study complies with the Guidelines for Reporting Reliability and Agreement Studies (GRRAS) (*Kottner et al., 2011*) and follows the taxonomy, terminology, and definitions from the "COnsensus-based Standards for the selection of health status Measurement INstruments" (the COSMIN study) (*Mokkink et al., 2010a*, *2010b*).

The choice of sample size, inclusion and exclusion criteria, and treadmill speeds was guided by the study aim and informed by previous study designs; the latter to facilitate comparison with other research in this field.

We find it important to investigate the criterion validity in healthy adults under controlled conditions prior to commencing a study in free-living conditions or in a non-healthy population.

| Table 1 Demographical data. | |
|---|---|
| Gender (N, men/women) | 12/18 |
| Age (years, mean (SD)) | 26.6 (±6.2) |
| Height (cm, mean (SD)) | 173.9 (±10.7) |
| Weight (kg, mean (SD)) | 73.8 (±12.4) |

## Ethics statement

The North Denmark Region Committee on Health Research Ethics stated that according to ethics committee law number 593 of 14/6/2011 § 2, 1-3 and § 14, 1 (Confirmation date 02.16.2016), ethical approval was not required for this study. The study was conducted in accordance with the principles of the Declaration of Helsinki.

## Sample size and method

The sample size was set to 30 participants, which was in accordance with equivalent laboratory condition validation studies (*Takacs et al., 2014*; *Kooiman et al., 2015*; *Riel et al., 2016*). Two observers conducted data collection, manual step count, data processing, and statistical analysis. The observers were blinded to the results until all raw data had been processed and were ready for statistical analysis.

## Participants

Thirty healthy adults (*n* = 18 females, *n* = 12 males), Table 1, were recruited through the University College of Northern Denmark by using a mailing list with contact addresses of students and staff. Verbal and written information on the study procedures and aim was provided, and written informed consent was obtained from all participants prior to commencing the study. All participation was voluntary. Data collection was conducted in April 2016.

Only healthy participants were included. The inclusion and exclusion criteria were in line with those of previous studies (*Hendrick et al., 2010*; *Steeves et al., 2011*; *Bergman et al., 2012*; *Fortune et al., 2014*, *2015*; *Takacs et al., 2014*; *Johnson et al., 2015*; *Stansfield, Hajarnis & Sudarshan, 2015*; *Hickey et al., 2015*). To ensure physical activity participation clearance participants were eligible for inclusion if they were: (i) aged 18–64 years (self-reported), (ii) able to read and comprehend Danish and English (self-reported), and (iii) responded "No" to all questions (7 out of 7) in the Physical Activity Readiness Questionnaire (PAR-Q) (*Shephard, 1988*; *Steeves et al., 2011*; *Johnson et al., 2015*). Moreover, (iv) for safety reasons, participants should be able to walk independently >40 min (self-reported) to ensure that they were able to complete the treadmill walking exercise without risking adverse events (e.g., injury). Criteria for exclusion were: (i) neurological diseases (self-reported), (ii) cognitive problems (self-reported), (iii) any musculoskeletal injuries and/or operation (self-reported) that might have affected walking six weeks prior to participation, and (iv) vascular issues (self-reported) since these may influence walking ability (*Takacs et al., 2014*; *Stansfield, Hajarnis & Sudarshan, 2015*; *Fortune et al., 2015*).

### Garmin Vivosmart® HR

The Garmin Vivosmart® HR (Garmin, Olathe, Kansas, USA: 149,99 USD (*Garmin, 2016a*) is a wrist-worn, accelerometer-based off-the-shelf activity tracker monitoring step count, distance, calories, heart rate, number of floors climbed and intensity minutes. The Garmin Vivosmart® HR has a build-in display for direct operation and reading of data. Data storage and analysis from the Garmin Vivosmart® HR can be assessed by the user through a Garmin Connect™ account enabling optional settings (e.g., Health & Fitness, Units). (*Garmin, 2016b*). To configure the Garmin Vivosmart® HR through Garmin Express™ v. 4.1.19.0, a Garmin Connect™ account was created with the Health & Fitness profile enabled. The Garmin Connect™ account was solely used to login to the Garmin Express™ v. 4.1.19.0 software to configure the activity tracker for each participant individually (*Garmin, 2016b*, *2016c*). The Garmin Vivosmart® HR was configured individually for each participant by entering his or her date of birth, gender, weight, and height. Metric values were used during this configuration process, time was presented in 24-h format, and the lowest activity level (standard value) was chosen for all participants. The standard option for the wearing side for the activity tracker was set to "left" in the Garmin Express™ v. 4.1.19.0 software. The Garmin Vivosmart® HR was placed proximally from the caput ulnae sinister on each participant, as recommended by the manufacturer (Fig. 1) (*Garmin, 2016c*).

### StepWatch™ 3

The StepWatch™ 3 (Modus Health, LLC, Edmonds, WA, USA: 525 USD (Accelerometer), 1470 USD (Docking station and software) (*Tudor-Locke et al., 2011a*)) is an ankle-worn, accelerometer-based, research-graded activity tracker monitoring step count and activity levels based on step rate. A docking station and software are needed to configure the StepWatch™ 3 and access the recorded data (*Tudor-Locke et al., 2011a*; *Modus Health, 2014*). The StepWatch™ 3 was configured individually for each participant by entering gender, height, weight, and age. Furthermore, the recording interval was defined to 15 s and Client ID and Trial ID were entered for subsequent data analysis. The two StepWatch™ 3 monitors (one on each leg) were mounted according to the manufacturer's recommendation just proximal from the malleolus lateralis of the ankle (Fig. 2) (*Modus Health, 2014*).

### Video recording and manual step count

Manual step count is regarded the gold standard for step counting during treadmill walking in laboratory conditions (*Tudor-Locke et al., 2002*; *Fortune et al., 2014*, *2015*; *Lee et al., 2015*; *Stansfield, Hajarnis & Sudarshan, 2015*; *Sellers et al., 2016*). Thus, manual step count was used during walking on a zebris FDM-T treadmill (zebris Medical GmbH, Isny, Germany). In this study, a GoPro Hero 4 Black Edition (GoPro, Inc., San Mateo, CA, USA) was used to record treadmill walking. The camera was placed so that the participant's sagittal plane was recorded.

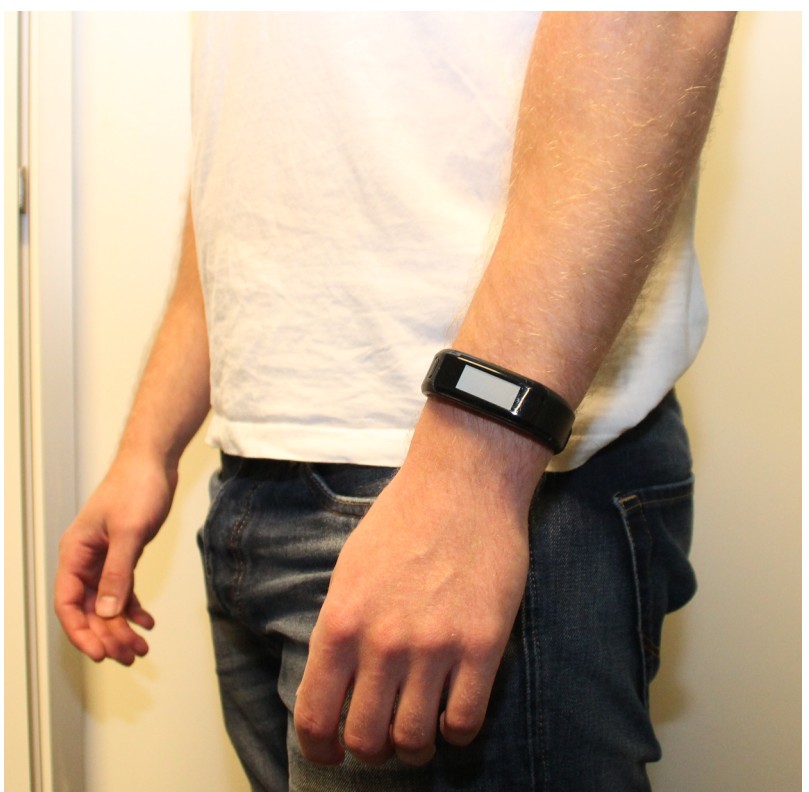

**Figure 1 Illustration of the placement of Garmin Vivosmart® HR.**

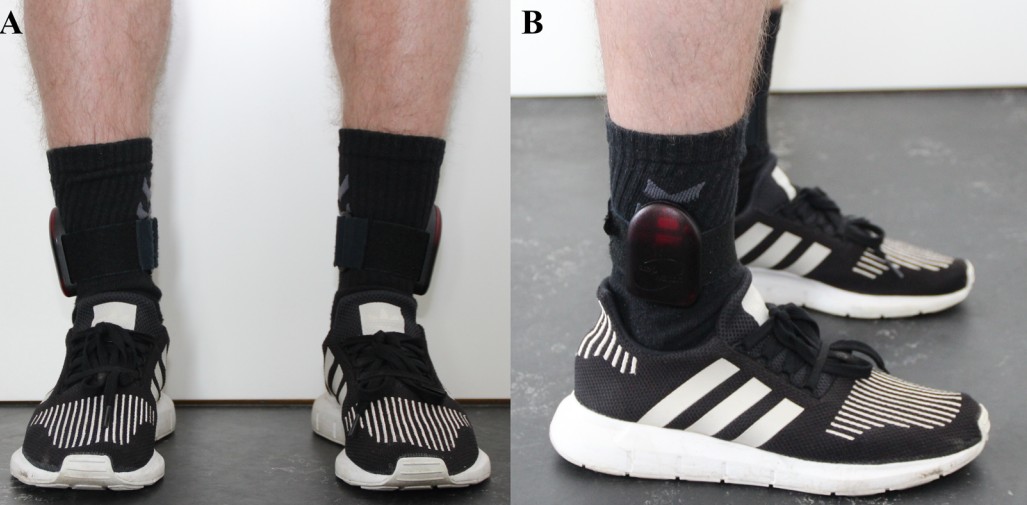

**Figure 2 Illustration of the placement of StepWatch™ 3 monitors.** (A) Frontal view, (B) Sagittal view.

In this study, a step was defined in accordance with Hickey and colleagues as "when the foot was completely raised off and subsequently lowered to the ground" (*Hickey et al., 2015*). Using this definition, the two observers individually viewed the recordings using

VLC version 2.2.4 (VideoLAN, Paris, France) while using a hand tally (Rucanor BSI ISO 09001 Certified No. FM40047) ensuring blinding and double validation of the manual step counting based on the approach in similar studies (*Rosenkranz, Rosenkranz & Weber, 2011*; *Aminian & Hinckson, 2012*; *Takacs et al., 2014*; *Riel et al., 2016*). In cases of discrepancies between the two observers' step counts, a new counting was performed to ensure valid counts as done by *Takacs et al. (2014)*.

## Procedure of the treadmill test

In order to examine the criterion validity of Garmin Vivosmart® HR and StepWatch™ 3 in step counting under controlled conditions, the participants were asked to perform six trials with a total sum of 40 minutes of treadmill walking at various speeds. Thus, to capture both very low, low, moderate, and high walking speeds, paces of 1.6 km/h, 2.4 km/h, 3.2 km/h, 4.0 km/h, 4.8 km/h, and 5.6 km/h were chosen, based on the approaches of previous studies (*Edbrooke et al., 2012*; *De Cocker et al., 2012*; *Lee et al., 2015*; *Gaz et al., 2018*). However, none of these studies included walking speeds below 2.4 km/h. We therefore found it particularly relevant to include the 1.6 km/h walking speed in this study as the use of activity trackers in healthcare may include target populations with intermittent decreases in walking speed (e.g., rehabilitation after ligament tears) or continued slow walking speeds (e.g., older or disabled people). In line with previous studies, each participant familiarized him or herself with the equipment before the actual test (*Ryan et al., 2006*; *Giannakidou et al., 2012*), walking the six different speeds for 2 min each to gain experience with the acceleration and deceleration of the treadmill.

As suggested in previous studies, a test period of 5 min per speed was used (*Le Masurier & Tudor-Locke, 2003*; *Crouter et al., 2003*; *Ryan et al., 2006*; *Giannakidou et al., 2012*; *De Cocker et al., 2012*; *Takacs et al., 2014*). An interval time break of 90 s was used to allow time to record data from the Garmin Vivosmart® HR and make a break in the recording of the StepWatch™ 3. The procedure for the interval time break was based on previous treadmill walking studies (*Bergman et al., 2012*; *Lee et al., 2015*). Increases and decreases in speed in each trial were programed to 0.91 m/s and were included in the 5-min test period. For safety reasons, verbal countdown from the observer was given 10, 5, and 3 s prior to the starting and stopping event. The treadmill was set to an elevation of 0 degrees verified by a spirit level, in line with similar studies (*Le Masurier & Tudor-Locke, 2003*; *Steeves et al., 2011*). As recommended by the manufacturer, service and calibration of the treadmill's pressure sensors and speedometer were performed prior to this study by an employe at *zebris Medical GmbH (2015)*. The tests were carried out in the Movement Laboratory, Department of Physiotherapy, University College of Northern Denmark, Aalborg, Denmark.

## Processing data

Garmin Vivosmart® HR: Start and stop values of step count recorded by the activity tracker were collected manually before and after every trial and transferred into a

Microsoft Office Excel Datasheet. The actual number of steps recorded by the activity tracker was calculated by subtracting the start value from the stop value.

StepWatch™ 3: When the participant had completed all trials, data were downloaded for both StepWatch™ 3 activity trackers with software provided by the manufacturer— StepWatch™ v.3.4. Data were exported with the software StepWatch™ v.3.4 into a Microsoft Office Excel Datasheet, where the sum of recorded steps by the two StepWatch™ 3 activity trackers was identified for each trial.

Video data from the GoPro Hero 4 Black Edition were transferred using the associated application/software GoPro Desktop v1.3.0.2371, available from the manufacturer's web page (*GoPro, 2016*). Data from each walking speed were registered using a visual speed indicator and saved as separate files for subsequent identification. The video recordings were used as the underlying technical basis for manual step count.

## Statistical analysis

All statistical analyses were conducted in *R* version 3.6.1 and *RStudio* version 1.2.1335. The criterion validity of Garmin Vivosmart® HR and StepWatch™ 3 was established by a series of statistical tests applying a significance level of $p < 0.05$.

The Shapiro–Wilk test was used to determine the distribution of normality of the differences between Garmin Vivosmart® HR and manual step count, and between StepWatch™ 3 and manual step count. The Shapiro–Wilk test showed events of non-normality ($p < 0.05$); therefore, non-parametric tests were applied for further statistical analysis. Medians and inter-quartile ranges (IQR) were calculated by the default settings in *R* for steps and differences in steps between manually counted steps and the two devices.

In order to assess the clinical relevance of the tested devices, the mean absolute percentage error (MAPE) was determined at each walking speed for Garmin Vivosmart® HR and StepWatch™ 3. The MAPE was calculated as $\text{MAPE} = \frac{|\text{Activity tracker} - \text{Manual Count}|}{\text{Manual Count}} \times 100\%$. Earlier studies have described a MAPE ≤3% to be clinically irrelevant (*Holbrook, Barreira & Kang, 2009*; *Colley et al., 2013*; *Johnson et al., 2015*; *Liu et al., 2015*; *Riel et al., 2016*; *Bunn et al., 2018*). The differences between manually counted steps and steps recorded by the two devices were presented in Bland Altman style plots along with the median, the 2.5th, and the 97.5th quartile due to the non-normally distributed dataset (*Gialamas et al., 2010*; *Riel et al., 2016*). All data were plotted, and no potential outliers were removed.

## RESULTS

### Garmin Vivosmart® HR vs. manual step count

The medians of differences in steps between Garmin Vivosmart® HR and manual step count were −49.5 (IQR = 101) at 1.6 km/h, −7.5 (IQR = 14) at 2.4 km/h, −2.5 (IQR = 6) at 3.2 km/h, −1 (IQR = 4) at 4.0 km/h, −1 (IQR = 5) at 5.8 km/h, and −4 (IQR = 7) at 5.6 km/h (Table 2). These are plotted in the Bland–Altman style plot (Fig. 3).

The results of the MAPE showed that the steps counted by Garmin Vivosmart® HR had clinically relevant deviations at 1.6 km/h, 2.4 km/h, and 5.6 km/h (MAPE: 3.49–26.35%) compared with manual step count (Table 2). The differences in step count

**Table 2 The median of counted steps by device, and the median of differences in steps by device compared to manual step count, and the mean absolute percentage error by device compared to manual step count.**

| Device | Walking speed (km/h) | Median (steps) | Median of difference (steps) | Mean absolute percentage error (%) ± SD |
|---|---|---|---|---|
| Manual step count | 1.6 | 337.5 (IQR = 33) | – | – |
| | 2.4 | 410 (IQR = 29) | – | – |
| | 3.2 | 473 (IQR = 38) | – | – |
| | 4.0 | 529.5 (IQR = 40) | – | – |
| | 4.8 | 574 (IQR = 46) | – | – |
| | 5.6 | 608 (IQR = 49) | – | – |
| Garmin Vivosmart® HR | 1.6 | 292.5 (IQR = 106) | −49.5 (IQR = 101) | 26.35 (±23.63) |
| | 2.4 | 398 (IQR = 34) | −7.5 (IQR = 14) | 3.49 (±4.10) |
| | 3.2 | 465 (IQR = 36) | −2.5 (IQR = 6) | 1.27 (±1.67) |
| | 4.0 | 527 (IQR = 37) | −1 (IQR = 4) | 0.61 (±0.61) |
| | 4.8 | 570 (IQR = 45) | −1 (IQR = 5) | 0.61 (±0.68) |
| | 5.6 | 577 (IQR = 53) | −4 (IQR = 47) | 6.45 (±11.46) |
| StepWatch™ 3 | 1.6 | 345 (IQR = 22) | 4 (IQR = 14) | 3.60 (±6.03) |
| | 2.4 | 412 (IQR = 29) | 0 (IQR = 1) | 0.35 (±1.00) |
| | 3.2 | 473.5 (IQR = 36) | 0 (IQR = 1) | 0.09 (±0.12) |
| | 4.0 | 529.5 (IQR = 40) | 0 (IQR = 1) | 0.08 (±0.09) |
| | 4.8 | 574 (IQR = 47) | 0 (IQR = 1) | 0.08 (±0.10) |
| | 5.6 | 608.5 (IQR = 49) | 0 (IQR = 1) | 0.09 (±0.12) |

for the device were interpreted as being clinically irrelevant as the values were below 3% at walking speeds of 3.2–4.8 km/h (MAPE: 0.61–1.27%).

## StepWatch™ 3 vs. manual step count

The medians of differences in steps between StepWatch™ 3 and manual step count were 4 (IQR = 14) at 1.6 km/h and 0 (IQR = 1) at 2.4 km/h, 3.2 km/h, 4.0 km/h, 4.8 km/h, and 5.6 km/h (Table 2). These are plotted in the Bland–Altman style plot (Fig. 4).

The results of the MAPE showed that the steps counted by StepWatch™ 3 had a clinically relevant deviation at 1.6 km/h (MAPE: 3.60%) compared with manual step count (Table 2). The differences in step count for the device were interpreted clinically irrelevant as the values were below 3% at walking speeds of 2.4–5.6 km/h (MAPE: 0.08–0.35%).

## DISCUSSION

This study aimed to investigate the criterion validity of Garmin Vivosmart® HR and StepWatch™ 3 at different walking speeds under controlled conditions. The results showed a tendency towards Garmin Vivosmart® HR undercounting steps compared with manual step count, and StepWatch™ 3 slightly overcounted steps compared with manual step count.
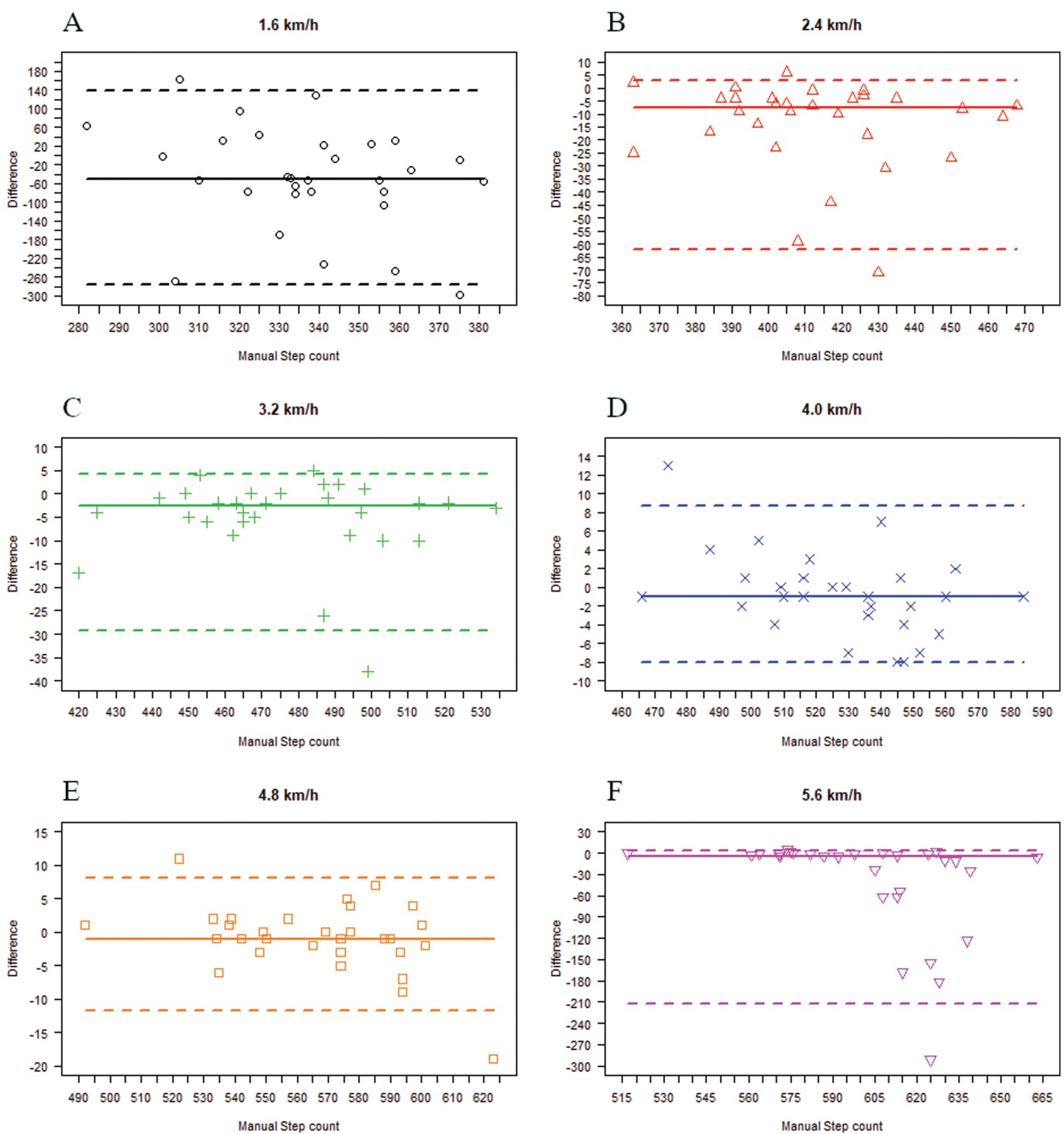

**Figure 3 Bland Altman style plot illustrating the differences between Garmin Vivosmart® HR and manual step count at walking speeds from 1.6 km/h to 5.6 km/h.** (A) 1.6 km/h, (B) 2.4 km/h, (C) 3.2 km/h, (D) 4.0 km/h, (E) 4.8 km/h, (F) 5.6 km/h.

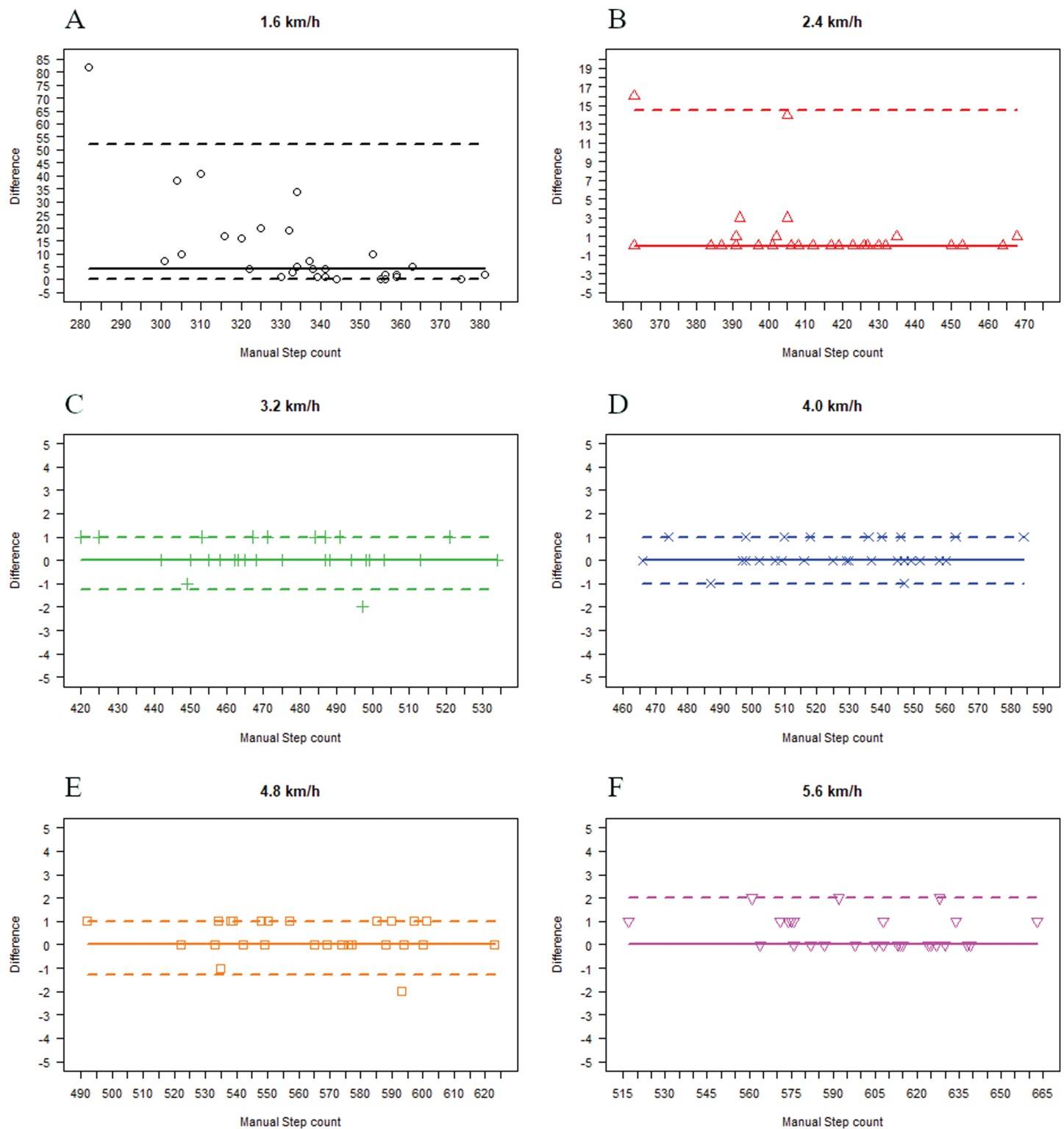

**Figure 4  Bland Altman style plot illustrating the differences between StepWatch™ 3 and manual step count at walking speeds from 1.6 km/h to 5.6 km/h.** (A) 1.6 km/h, (B) 2.4 km/h, (C) 3.2 km/h, (D) 4.0 km/h, (E) 4.8 km/h, (F) 5.6 km/h.

Despite their tendencies to undercount and overcount steps, the activity trackers may still be clinically relevant at some walking speeds when a cut point of a MAPE ≤3% is used (*Holbrook, Barreira & Kang, 2009*; *Colley et al., 2013*; *Johnson et al., 2015*; *Liu et al., 2015*; *Riel et al., 2016*). The results showed that both Garmin Vivosmart® HR and StepWatch™ 3 miscounted steps at slow walking speeds, and that Garmin Vivosmart® HR miscounted steps at fast walking speeds as well. Taking this into account, we find that Garmin Vivosmart® HR is valid for counting steps at walking speeds from 3.2 km/h to 4.8 km/h and close to the cut point at walking speed of 2.4 km/h with a MAPE of 3.49%. If the aim is to walk 10,000 steps per day, Garmin Vivosmart® HR would miscount 349 daily steps at a walking speed of 2.4 km/h. However, the MAPE also shows that Garmin Vivosmart® HR is far from valid when counting steps at very slow and fast walking speeds. Old people tend to walk slower than younger healthy people and many also reduce their physical activity level (*Burton et al., 2018*). Accurate measuring of walking speed and amount of walking is important because both speed and amount of walking are associated with health outcome in older people (*Graham et al., 2010*; *Lee et al., 2019*). However, at a slow walking speed of 1.6 km/h, Garmin Vivosmart® HR would miscount around ¼ of all steps, which makes it highly unreliable in a clinical setting with patient groups walking at slow speeds.

When assessing MAPE for StepWatch™ 3, we found that this device is more valid in detecting steps than Garmin Vivosmart® HR at all walking speeds. StepWatch™ 3 only exceeds the cut point slightly at a walking speed of 1.6 km/h with a MAPE of 3.60%. This supports previous findings stating that StepWatch™ is one of the most accurate activity trackers for step counting at slow walking speeds (*Lim et al., 2018*). Both devices showed difficulties in counting steps, especially at slow walking speeds, which supports the results of previous studies of activity trackers (*Crouter et al., 2003*; *Steeves et al., 2011*; *Van Remoortel et al., 2012*; *Fortune et al., 2014*; *Fokkema et al., 2017*; *Bunn et al., 2018*; *Gaz et al., 2018*). A plausible explanation suggested in previous studies of other activity trackers may be that the algorithm used for step detection at slow walking speeds was insufficient (*Taraldsen et al., 2011*; *Fortune et al., 2014*) and that very slow walking therefore may not generate sufficient acceleration for registration by accelerometery (*Fokkema et al., 2017*; *Gaz et al., 2018*). It is a challenge that the step-detection algorithm is a closed source in most consumer activity trackers because this prevents analysis of how the accelerometer raw data are processed. Another general issue is that some activity trackers are updated or become outdated before validation studies are published (*Bunn et al., 2018*). It should be noted that Garmin Vivosmart® HR has been discontinued (*Garmin, 2020*). Garmin's newest Vivosmart® is the Garmin Vivosmart® 4, and due to the closed source firmware, it is unknown but also unlikely that newer devices are programed with the same firmware as older models. In general, it should be mentioned that manufacturers frequently update their software. Ongoing optimization and perfection of algorithms are likely to affect activity detection and thereby step counts. Therefore, the results of this study should be interpreted with consideration to the software versions at the time of data collection; 18–27th of April 2016.

The different placements of the activity trackers may also affect their ability to detect steps. Anderson and colleagues found that the placement of the accelerometer affects the criterion validity; that is, ankle-mounted devices seem valid, whereas thigh-worn devices are not considered valid using ActiGraph GT3X for step counting among ward-based adults (*Anderson, Yoward & Green, 2019*). In addition, studies of hip-based activity trackers have found that these devices provide more accurate step counts than wrist-based trackers (*Gaz et al., 2018*). Due to its placement on the wrist, the accelerometer in Garmin Vivosmart® HR will primarily be affected by arm swing and upper body movement, whereas the accelerometer in StepWatch™ 3 is more likely to be affected by shuffling, stride length, etc., due to its placement on the ankle.

As mentioned previously, manual step count is considered the gold standard in laboratory conditions, but measuring daily steps in free-living conditions by manual step count would be very impractical. Results indicate that the research-graded StepWatch™ 3 is the best device for measuring step count in free-living conditions due to its accuracy at both slow and fast walking speeds. However, the cheaper, consumer-graded Garmin Vivosmart® HR can be used in a target population with a narrower range of walking speeds, from relatively slow to moderately fast walking speeds (3.2–4.8 km/h). If the target population meets the criteria for walking speeds of Garmin Vivosmart® HR, using this device could be a cost-benefit-balanced approach to measuring steps. This information could aid decision makers and healthcare personnel when considering to implement physical activity trackers in clinical practice and provide an indication on whether they might consider using a consumer-graded activity tracker instead of a research-graded activity tracker in their target population in relation to walking speed.

## Strengths and limitations

The use of different walking speeds under controlled conditions in general and the inclusion of walking speed down to 1.6 km/h in particular are considered a strength of the present study since this has not previously been addressed sufficiently in criterion validity studies. Very slow walking speeds are typically found in older or disabled people; however, the results of the present study cannot be directly compared to older people as only healthy younger adults participated in our study. The gait pattern in younger people may differ from the gait pattern in older people (*Burton et al., 2018*). The present study only investigated step detection under controlled conditions in straight treadmill walking, and it should be noted that sensitivity and specificity may be very different in free-living conditions and in people with disabilities and alternate walking patterns, for instance.

This study adds to the knowledge base on criterion validity of step count under controlled conditions for consumer and research-graded activity trackers. Even though this study investigates the criterion validity of two specific activity trackers, the findings may guide decision makers and healthcare personnel's choice when implementing and applying activity trackers in clinical practice.

### Future work

Future studies should validate and compare consumer and research-graded activity trackers in free-living conditions and include non-healthy participants whose movement patterns and mobility disorders might influence the ability of the activity trackers to detect steps (*Steeves et al., 2011*; *Stansfield, Hajarnis & Sudarshan, 2015*; *Fokkema et al., 2017*; *Burton et al., 2018*). Furthermore, it would be relevant to investigate the ability of the activity trackers to detect steps and reject non-steps in activities such as getting in/out of bed, sitting and raising from a chair, jumping, etc., to expand knowledge on the ability of the activity trackers to discriminate steps from other free-living activities in different populations.

## CONCLUSION

This study concludes that both the consumer-graded off-the-shelf activity tracker (Garmin Vivosmart® HR) and the research-graded activity tracker (StepWatch™ 3) are valid in detecting steps at selected walking speeds in healthy adults under controlled conditions. However, both activity trackers miscount steps at slow walking speeds, and the consumer-graded off-the-shelf activity tracker also miscounts steps at fast walking speeds. Even though the results reflect the criterion validity of two specific activity trackers under controlled conditions, the findings may inform and support decision makers and healthcare personnel when implementing physical activity trackers and deciding whether to use a consumer-graded or a research-graded activity tracker.

### Funding

The authors received no funding for this work.

### Competing Interests

The authors declare that they have no competing interests.

### Author Contributions

- Frederik Rose Svarre conceived and designed the experiments, performed the experiments, analyzed the data, prepared figures and/or tables, authored or reviewed drafts of the paper, and approved the final draft.
- Mads Møller Jensen conceived and designed the experiments, performed the experiments, authored or reviewed drafts of the paper, and approved the final draft.
- Josephine Nielsen conceived and designed the experiments, authored or reviewed drafts of the paper, and approved the final draft.
- Morten Villumsen analyzed the data, authored or reviewed drafts of the paper, and approved the final draft.

## Human Ethics

The following information was supplied relating to ethical approvals (i.e., approving body and any reference numbers):

The North Denmark Region Committee on Health Research Ethics stated that ethical approval was not required for this study (Confirmation date 02.16.2016).

## Data Availability

Raw data and code are available in the Supplemental Files.

## Supplemental Information

Supplemental information for this article can be found online at http://dx.doi.org/10.7717/peerj.9381#supplemental-information.

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
