# Peer review of "The validity of activity trackers is affected by walking speed: the criterion validity of Garmin Vivosmart® HR and StepWatch™ 3 for measuring steps at various walking speeds under controlled conditions"

_PeerJ, doi:10.7717/peerj.9381_

## Round 0.1 · original submission · Major Revisions

While there is some variation in opinion between two reviewers, I feel that you deserve the opportunity to respond to these comments. In particular, please be aware of reviewer one's comments when making your response to reviewers comments letter and revisions to the manuscript. Particular emphasis needs to be placed on updating the literature reviewed in the introduction and answering the query regarding a number of aspects of the methodology including the application this data to older adults, even though younger adults were the sample utilised.

·

Basic reporting

This paper compares step counts from two wearable accelerometers (Garmin Vivosmart HR and StepWatch 3) with observer-counted steps during treadmill walking. The trial was repeated six times, each with a different walking speed (1.6–5.6 km/h). I found the paper interesting to read and it is reasonably well written. There are several small errors with wording or sentence structure throughout. While these errors are small, they collectively reduce the quality of the work. I suggest a thorough proofread.

The field of physical behaviour measurement has evolved rapidly over the last several years. I find it odd that the authors do not acknowledge any recent research – including research which specifically examines the validity of the two devices assessed in this study. After a quick search, I see 5+ papers published in the last four years examining the validity/reliability of these two wearables. This paper seems as if it was written four years ago and not updated (given the most recent reference is 4 years old). I suggest the authors reposition the rationale considering recent evidence (i.e. position it such the rationale for replication is clear).

The layout of the paper and results tables are presented well. A small suggestion would be to drop Table 3 and simply add an additional column to Table 2. For the figures, there is duplication (i.e. same data are shown in Figures 5, 6, and 7). I would suggest collapsing Figures 3–7 into two new figures: one for the Vivosmart and one for the StepWatch. Each figure would have 6 panels/facets for the different walking speeds. This would make it easier to visualise the mean difference and LOA for each device/speed combination.

Experimental design

The overall design seems reasonably sound (see some specific limitations below). I understand the general purpose of the study, but the written aim (at the end of the introduction) is not well articulated. This study “aims at creating a knowledge base…inform…enlighten…” is not a measurable aim. I suggest rephrasing this, so it is specific and measurable.

What you are primarily assessing in this study is the accuracy of the manufacturer’s step detection algorithm(s). One of the main issues with consumer-grade wearables is the activity detection algorithms are closed-source (and black box). This also means that the manufacturer could update these algorithms without the user knowing. I think one of the main study design ‘issues’ is the use of the Garmin Vivosmart HR. This device was discontinued in Q1 2016 (Garmin only produced it for ~6 months). It is very unlikely the step detection algorithm in the HR would be identical to the one used in the current Vivosmart 4. This issue has not been discussed.

The rationale for many of the design decisions (e.g. sample size, inclusion criteria, treadmill speeds) is justified ‘based on previous work’. This is fine, but I feel these decisions should also be rationalised within the context of the current study.

One of the main rationale points in the introduction is the slow walking speeds in elderly. The sample consisted of 30 healthy adults, which limits the generalisability to other populations, particularly elderly. Slow walking speeds alone do not account for the biomechanical differences in walking. Step frequency could be similar but stride length much shorter in elderly (for example). This limitation is not described (in fact, no limitations are presented in the discussion).

In my mind, accurate step detection consists of two parts: 1) the ability to detect steps, and 2) the ability to reject non-steps. The current study design only addresses the former, meaning you can only assess the sensitivity of each step detection algorithm and not the specificity. A protocol that includes things like shuffling, standing with movement, jumping, kicking a ball (ankle sensor), throwing (wrist sensor), etc. would be needed to demonstrate specificity.

Validity of the findings

All underlying data and analysis code have been provided, which is excellent, and makes the results reproducible. However, I feel the conclusions of the paper are somewhat overstated. Stating that ‘This study concludes that the Vivosmart and StepWatch are valid in detecting steps…’ is slightly misleading. The conclusions should be presented in the context of the limitations mentioned above, and the existing work which was not considered.

Additional comments

Overall the study was reasonably well-conducted, and the paper was enjoyable to read. The main issue is not recognising the limitations of the design, and not considering the (substantial) new evidence that has been produced over the last 4 years.

Reviewer 2 ·

Basic reporting

1. There are sections where the english can be improved. For example, in the first paragraph, the frequent use of “e.g” is incorrect.

Lines 61-74, the authors switch back and forth from past to present tense. This makes it confusing for the reader.

2. Line 4- Recent work by iMin lee found this 10,000 step recommendation to be misleading. The authors should update this section to highlight recent work done examining the validity of these recommendations.

Experimental design

The authors need to add additional information into the introduction explaining the the differences in the step watch vs Garmin. Wrist worn vs ankle worn. Another paragraph discussing the various locations and how they can impact device accuracy would benefit the reader.

The authors state that “ethical approval” was not required? But then the participants signed an informed consent? Was IRB approval needed or not? If not, why did they have the participants fill out the Par Q and other materials? Clarity here would be appreciated.

Validity of the findings

Well designed study.

I would like the authors to add more into the discussion as to how these devices could be used among a clinical setting, seeing as they introduced that possibility in their introduction.

Additional comments

I enjoyed reading this manuscript. I believe it offers value to the field and will assist future investigators determine whether these devices should be used in their research.

Grammar needs to be cleaned up.

---

## Round 0.2 · accepted · Accept

I wish to thank you on behalf of the reviewers and the journal for responding adequately to the reviewers' initial concerns. I, therefore, recommend this manuscript is accepted for publication in PeerJ.

·

Basic reporting

no comment

Experimental design

no comment

Validity of the findings

no comment

Additional comments

I'm happy with the authors' (substantial) revisions and rebuttal responses. The paper is much better than the initial manuscript, and probably warrants publication in PeerJ.

Reviewer 2 ·

Basic reporting

I'm satisfied with the authors' response to my suggestions.

Experimental design

I'm satisfied with the authors' response to my suggestions.

Validity of the findings

I'm satisfied with the authors' response to my suggestions.

Additional comments

Nice job responding to both reviewers' comments/edits. I look forward to seeing this in print.